# Psychomedical Interventions with Transgender People in Portugal and Brazil: A Critical Approach

**DOI:** 10.3390/ijerph19010267

**Published:** 2021-12-27

**Authors:** Liliana Rodrigues, Matilde Soares, Conceição Nogueira

**Affiliations:** Centre for Psychology, Faculty of Psychology and Education Sciences, University of Porto, 4200-135 Porto, Portugal; matildesoaresbc@hotmail.com (M.S.); cnogueira@fpce.up.pt (C.N.)

**Keywords:** psycho/medical interventions, transgender, critical psychology

## Abstract

This study aims to analyze biopsychomedical interventions with transgender people. For this purpose, we carried out 35 semi-structured interviews with people who self-identify as transsexuals and transvestites in Brazil and Portugal. The responses of the study participants were systematized according to a thematic analysis, which led to the emergence of the following three main themes: “institutional power”, “expectations of trans-bodies”, and “experiences in health services”. This study demonstrates how some trans people perform bodily modifications to fight the transphobia they experience throughout their lives. In addition, they believe that, by making their bodies conform to each other, they may become more attractive and desirable. The process of cisnormativity is, furthermore, conveyed by the idea present in the answers of some respondents: that having “integrated” bodies means facing less discrimination and that they will, therefore, obtain more satisfactory ways of personally and socially experiencing their identities. This study contributes to a deepening critical reflection on the experiences/exclusions of trans people, especially in the psychomedical context of “normalization” devices. Hence, just as social structures produce and sustain transphobia, the same structures are responsible for combating it.

## 1. Introduction

In Portugal, on 9 September 2019, in the city of Almada, Lara Crespo, a Portuguese trans woman committed suicide. Lara, similar to Gisberta Salce Júnior, was not only a trans woman but also 48 years old and precarious in nature. Living her life on the margins, Lara was constantly exposed to hatred and, consequently, endured extreme vulnerability. Lara committed suicide; Gisberta was murdered [1]. Both were targets of the same violence: transphobia.

As a worldwide problem, transphobia is not treated with due attention. Some studies contributed to the social stigmatization of transgender people (e.g., [2,3]) by locating the problem in transsexuals and not questioning or problematizing transphobia. Therefore, negative referential frameworks of intervention were produced that, instead of accompanying individuals in making decisions about their identities, served as the gatekeepers of the sex/gender systems [4,5,6,7,8,9,10].

When the sex–gender congruence is not verified, these subjects are often diagnosed as having “gender dysphoria” [11] and/or “gender incongruence” [12], which, despite no longer being considered a pathology in the latest version of the World Health Organization’s International Classification of Diseases, continues to be present as a medical condition, thus maintaining control and pathologizing these identities. Some of the diagnostic criteria involve and are based on the discomfort with one’s own primary and/or secondary sexual characters, the desire to free themselves from them, and the accentuated desire for the primary and/or secondary sexual characters of the “other” gender, different from the one designated at birth [11]. These discourses impose, then, the desire for physical changes as the only “way out” of this “non-conformity” and as necessary to identify themselves as trans, demarcating the insistence on seeing gender as having a biological basis [7,9,10,13].

In fact, some societies are organized in such ways that exclude even conceiving of this idea of a non-compliant body, delegating the action of pathologizing and “disciplining” human experiences to the medical authorities, in this case of non-compliant bodies and genders [9,10,14].

Furthermore, according to the sociologist and trans activist, Miquel Missé [5], the trans body has been put forward as a paradigm of “error”, as if some problem happened at some point in that person’s life. This paradigm emphasizes how trans people were born into the wrong body but will fortunately be able to correct and recover from this situation. The “error” paradigm has harmed trans people far more than it has benefited them. Above all, this paradigm embodies the perspective that as people “feel” that their body is not their own, they tend to mistreat, mutilate and destroy it. Correspondingly, maintaining the paradigm of error, and the need to correct and recover it, clearly carries risks [5,6,9,15].

In addition to all the other issues involved in the process of pathologizing trans identities, these bodies were also symbolically designated as abject and “monstrous” bodies [16,17,18]. To be accepted, these bodies must go through “normalization devices”, with these devices being heteronormative. Maintaining the pathology in these bodies involves maintaining the idea that these bodies will only become desirable, and may indeed wish to go through, processes of heteronormative and cisnormative “disciplinization”, constituting a process of hierarchization and legitimization of bodies and lives according to these determined systems [9,10,14,16,19,20].

Furthermore, in order to access healthcare, some trans people may create the legitimate narratives of the medical model (e.g., expressing a desire to live as the other gender since childhood, and feelings of rejection toward their genitalia and being heterosexual) [5,9,10,20].

This study approaches depathologization as the potential scope for legitimizing the right for individuals to have autonomy over their bodies and identities, creating visible bodies and spaces and places of resistance [9,10,16,21], assuming that this premise entails not only problematizing the models but also human lives themselves [5]. As an example, Miquel Missé [5] states, “our bodies have no problems. The problem is the system that does not know in which drawer to place them, classify them, read them. But of course, the operating room is much more economical and less questioning than social change” (p. 71, free translation).

In this sense, the main objective of this study is to analyze the responses of trans people in Portugal and Brazil regarding their experiences in healthcare contexts, and thereby refine knowledge about the meanings they attribute to their life histories in these contexts and their relationships with their identity (de)constructions.

## 2. Materials and Methods

### 2.1. Participants

This study interviewed 35 individuals (none of which were the authors of this paper) who self-identify as transsexuals, transvestites and trans; these terms were attributed by the respondents themselves. “Tranvestite” is an identity and designation adopted in the Brazilian context (“travesti” in Portuguese). This designation is opposite to those called “transvestite” or “cross-dresser”, in a European context and, specifically, in Portugal. The designation “transvestite” in Portugal refers, with some consensus, to people who sporadically dress and express a gender opposite to that which they were assigned at birth but still identify with the gender assigned to them in the birth registration [5,6]. Despite the differences between these concepts—transexuals, transvestites, and trans—we apply the term trans unless respondents specify other designations, especially in the Section 2.4 and Section 3. The biographical data of the study participants also result from the analytical process that is itself shaped by the research paradigm(s). The biographical data were constructed in keeping with the reality of the study, privileging the self-designations of the people interviewed. When we collected the biographical data, we applied the identification categories that might best designate the study participants. Some categories were linked, allowing new analytical configurations.

Central to all the participants is the discourse of non-conformity between the sex designated on their birth certificates and the gender with which they identify, with a spectrum of possibilities for experiencing this same gender. Twenty-one people identified themselves as female gendered and fourteen as male gendered, with ages between 16 and 55 (M = 30.17 and PD = 8.75). Twenty-four of these participants were Brazilian, whereas eleven were Portuguese.

Thirty respondents were single, three were married and two divorced. Twenty-one participants identified themselves as heterosexual, five as bisexual, one person identified himself as gay, two defined themselves as lesbians, one declared a lack of physical, psychological or emotional attraction to anyone, four respondents stated an attraction to people, and one person said they did not know. In terms of their educational qualifications, two respondents finished basic education, twenty-one people completed secondary school and twelve hold higher education qualifications. Finally, regarding their professional situations, eleven people were in regular employment, four were unemployed, while twenty respondents were involved in other professional situations (e.g., precarious, non-working jobs, research fellows and sex workers).

With the concern of protecting the identity of the study subjects, an identification code based on the characteristics of these subjects was created. The identification code of a subject begins with the interview number, followed by the initial of their name, their chosen gender, the initials of the country and finally the initials of a large town or small town (in Portugal) or the initials of the region of the country to which they belong (in Brazil). For example, the code “17-B.M.BR.CO” corresponds to the interview number 17, the initial of the participant’s name is B, they are a masculine/trans man and are from Brazil, specifically the Midwest region. It should also be noted that, to further guarantee the confidentiality and anonymity of the participants, the letters chosen for their initials are random and, therefore, do not coincide with the names with which they identify themselves.

### 2.2. Instrument

We deployed a semi-structured interview script as a data collection tool for this study. As people from the historical, socio-cultural, economic and political contexts of Portugal and Brazil were interviewed, we produced two versions of the same interview guide (Portugal and Brazil). At the time that the data were collected, the legal framework of the two countries implied that there were questions that did not apply to both contexts, which led to the adaptation of the script. Moreover, despite the fact that the language of both countries is Portuguese, there was a need for cross-cultural adaptation of the language. The interview guide was drafted after previous consultation, and the subject was deepened by a review of the literature on trans issues.

This script contained three sections: the first referred to informed consent, in which the participants read the terms of study participation; the second relates to the interview itself, i.e., the semi-structured questions enabling us to respond to the purposes of this research; and, finally, the third was provided for the collection of the participant’s biographical data.

Regarding the interviews themselves, the script questions were grouped into the following four central themes: subjective identities, human rights, services and life paths. Regarding services (material especially relevant to the analysis in this article), we refer to the context of healthcare services to understand the perceptions and experiences of health services. In this group of questions, we sought to determine whether the perceptions the respondents had about their designations (e.g., transsexuals, transgender/transvestites) caused them to contemplate their right to healthcare; their perceptions about the requirements of some European countries for sterilization surgeries to be performed before an individual could change their name and sex in the civil registry, and how some individuals had to divorce their partner in order to have their legal identity recognized; whether the person had accessed any health services and if so, which types; their experiences with these services and specifically the work of any psychologists; and, finally, the kind of perceptions and experiences they encountered in real life.

Throughout the interview process, the themes described only served as general guidelines for conducting the interviews in order to assign some structure to them; however, they were always applied flexibly according to the interviewee and adapted according to the countries (i.e., Brazil and Portugal).

### 2.3. Procedure

The data collection process was intentional by design, as there was privileged contact prior to data collection with some study participants. The invitation for interviews and subsequent data collection covered two phases, first in Brazil (October 2013 to January 2014) and then in Portugal (March 2014 to October 2014).

The interviews were usually held in public places (chosen by the interview respondent) but in relatively empty places to ensure appropriate audio recording conditions, and lasted for an average duration of 60 min. After the full transcription of each interview, we carried out data analysis. NVivo 8.0 software was applied to organize the interview contents primarily due to the volume of material collected.

### 2.4. Data Analysis Assumptions: Constructionist and Feminist Thematic Analysis

This study adopted a constructionist, feminist and intersectional paradigm for the thematic analysis of the data. This analysis was more deductive (theoretical), in keeping with how the literature survey on the theme informed the practice. However, some problems emerged from the data; therefore, the analysis also took on a more inductive character that made it possible to strengthen the analysis. The themes encountered were semantic in the first phase and more latent (and more in-depth) in the second phase of analysis. We were able to identify more latent themes in participant responses through knowledge of the historical, socio-cultural, economic and political contexts of Portugal and Brazil. This knowledge was derived from the first author’s activism in both these countries, from the detailed reading approach made to the theme, for both countries, prior to the application of the interviews, and the privileged researcher contacts with the diverse contexts of proximity spanning the concrete lives of these people. The familiarity with these contexts enhanced the analysis of these data, allowing for a deeper understanding of the participant’s responses within the context they derived from.

In addition to the set of concepts and paradigms requiring definition, Braun and Clarke [22] also defined a set of phases or steps to be used when starting out on a thematic analysis process to guide the whole analytical process. By following these steps, we were able to obtain the results that are presented below. Furthermore, the various thematic maps produced in each phase of the analytical process provided a better understanding of both the analytical process carried out and the successive decisions on defining and redefining the themes, their sub-themes and codes.

## 3. Results and Discussion

This section presents the analysis emerging from the data, shaped by the respective constructionist, feminist and intersectional paradigms. From this analysis, we may identify three related themes, specifically the following: (i) institutional power, (ii) expectations of trans bodies and (iii) experiences of health services. The issues emerging from the data are analyzed and discussed in detail below.

### 3.1. Institutional Power

From the identification of the codes, one of the topics identified and defined was “Institutional Power”. The process of defining this theme aimed to analyze power relations in (bio)psychomedical contexts involving interventions with trans people. Three codes emerged from the data, namely, the “normalization of services”, “normalization of trans bodies” and “questioning the normalization of services and trans bodies”.

Nineteen participants mentioned that the (bio)psychomedical intervention process constitutes a process in which there is a “normalization” of services, meaning that several people must follow the same process, that is, the same duration before going from one service to the other, the same intervention phases and the same services. For example, while some people need a longer psychological follow-up than that stipulated by WPATH SOCs, others may feel that this period is too long for their well-being, and this prolonged duration may even worsen their life conditions. The following is an example:


*“She goes to the center, requests care from the social worker, imagine that she talks to the social worker, look, I’m a transsexual, this and that, the social worker will forward it to the psychologist, she has to go through the psychologist and there will be all that chatter (…), they will indicate her to an endocrine, and it is a process that takes a long time. And then until it culminates in the surgery, in the surgery row, actually, not in the surgery, the surgery row”.*
(3-F.F.BR.SE)

These data corroborate some studies in the area as they reinforce how these services are “normalized”, and thereby fail to consider the specificities of their own unique pathways [9,23,24,25]. Although, the WPATH SOCs, in their most current versions, do not reinforce a single intervention path for trans bodies, referring to a variety of therapeutic paths, in practice (according to these study participants). 

The other code, the “normalization of trans-bodies”, demonstrates the extent to which, in addition to services being homogenized and “normalized”, trans bodies are also homogenized as if there is only one way to construct themselves as trans within health services [9,10]. This “institutionalization of normalization” takes place in health services, is regulated by their families and “imposed” by other trans people in order to validate their bodies and identities. The following is an example:


*“I’m aware that it’s based on standardizing assumptions, ok, so if it’s T. now and it’s been recognized that it’s T., we want it to be T. all the way, and from there they could base their support for me to do the surgeries. (…) although in my daily life I can base trans issues on other assumptions, and I am claiming that I understand that my parents have these assumptions and accept me according to these assumptions. But they are complete, they support me 100% and they make a point of giving their love 100%. And they are ok with all this. And very proud of me”.*
(3-T.F.PT.PL)

Another code that emerged from the data was “questioning the normalization of services and trans bodies”. Many of the participants in this study questioned the impact of normalization devices on their bodies and, consequently, on their identities. Some literature reinforces the willingness of trans people to reproduce binary bodies; however, other studies question this same reproduction (and disciplining) of bodies [6]. This code effectively conveys how these people do not stop problematizing the impact of biomedical interventions on their bodies. This problematization did not arise only because many of those interviewed were involved in activism, it was also put forward by people who pervade various contexts and spaces of existence, and many questions arose precisely because some people took the contexts of (bio)psychomedical intervention into account. The following is an example:


*“(…) I know that each one has a desire, but I think it was a desire that was installed (…). I think it doesn’t exist unless the other has made it the norm (…). “All [trans woman] cannot touch your penis, all cannot, all had conflict, all from little ones”, (…) evidently there is a certain femininity, but not a rejection of the penis, even because it is an object of pleasure, in all the senses, and there are traumas that lead you to… it is not a child who has all the expertise of assembling these speeches, then one starts to really see things”.*
(1-PFBR.S)

As in other studies, the interviewee discourses demonstrate how the medical discourse provides the official and hegemonic discourse on trans life paths, restricting other possibilities of (re)constructing bodies and/or (de)identities [5,9,10]. In fact, some societies are organized in such a way that to conceive this idea of a non-conforming body is not acceptable, delegating to the medical power the action of pathologizing and “disciplining” human experiences, in this case, of non-conforming bodies and genders. The doctors, therefore, become gatekeepers of the sex/gender binary system [5]. This power is also anchored in the idea that the regulation of bodies has to accompany the regulation of (social) identities and, thus, the processes of body modification become the only way to maintain gender conformity [16]. In other words, the processes should follow gender norms (e.g., male sex–male gender and female sex–female gender) and heterosexual sexual orientation as the only possible and acceptable orientation [16,19].

To better illustrate the “institutional power” theme, the map of this theme and its codes is shown in Figure 1.

### 3.2. Expectations of Trans Bodies

The following three codes emerged from the data: “surgery (body modification) as identity legitimization”, “surgery (body modification) as a fantasy for happiness” and “surgery (not) as fate” that underpin the identification of the “expectations of trans bodies” theme.

Some participants convey how much of the desire for sexual reassignment surgery arises as a way to legitimize their identity to the “other” [9,10,26], as if it were not equally legitimate to recognize identity beyond the genitalia. To build themselves as women or men, to build themselves as people, extends beyond the genitalia or the sexual characteristics they possess. However, living in a binary, transphobic/sexist society may imply that some people transform their bodies, go through cisnormative “normalization devices” in order to recognize themselves as men or as women, to recognize themselves as people, as humans (and to be recognized by them) [9,10,27,28]. An example of the code “surgery (body modification) as legitimization of identity” follows:


*“(…) so I end up going back to what I said that I was deprived of a basic right simply because I don’t have the surgery. (…) no one is seeing my genitalia, but my face, my construction as a person, everyone is seeing it. And my construction as a person does not match that name, often with that identity photo, but people prefer to see what is covered than what is in their face”.*
(23-L.F.BR.S)

Another code that emerged was “surgery (body modification) as a fantasy for happiness”. From this code, we may analyze how some transgender people perform bodily modifications to combat the transphobia/sexism that they experience throughout their lives. Additionally, also anchored in transphobia, they believe that by making their bodies conform to each other, they shall be more desirable and attractive. The process of cisnormativity is then implemented through the recognition that, with redefined normative bodies, they will face less discrimination and, therefore, encounter more satisfactory ways to experience their identities personally and socially. This is reflected in the following interview excerpt:


*“You have to have surgery so that you feel complete and this completeness is pure fantasy that she will never have, neither with this surgery nor with any other surgery in the world or only if you were to take her brain out and she would live without memory”.*
(1-P.F.BR.S)

The code “surgery (not) as fate” identifies how not all trans people desire or undergo sexual reassignment surgery. From these data, we may analyze how surgery should not represent a requirement to legitimize identity. There is greater importance in problematizing just what leads people to undergo social reassignment surgery than highlighting the reasons that lead people not to have such surgery. Nevertheless, the desire and/or performance of sexual reassignment surgery should in no way constitute a requirement to legitimize any identity, as people do not share the same contexts of medical, social, political and historical resources. However, it must be recognized that this surgical procedure can serve as the means for people to live better by promoting their well-being (as oppressive systems continue to exist), but that not everyone needs surgery; they break the (binary) gender norms and (re)construct themselves as people, regardless of their genitalia. The following provides an illustrative extract:


*“If I had very good, quality surgery, I would do it. I always felt like having a penis, but I don’t want to have a defective penis. I don’t want to have a defective penis and I also have no problem with my genitalia. (…) I would just like to have a penis that works and that is pleasant for me and also for other people that I relate to. I never wanted something strange, let’s say, nothing that would mutilate me. You know, I don’t have the courage to do that”.*
(20-C.M.BR.S)

To better illustrate the theme of “Expectations of trans bodies”, Figure 2 sets out its map and codes.

### 3.3. Experiences in Health Services

Finally, the following codes emerged from the data: “non-transparency in the process and questioning of quality”, “slowness of services”, “real life experience(s) and their questioning”, “non-legitimation of identity by professionals”, “need for follow-up and non-evaluation”, “lack of information and training” and the “need for specialized services”, which collectively constitute the “experiences in health services” theme.

Some participants point out the lack of transparency in sexual reassignment processes and do so to the extent of even questioning their quality. The reference to the lack of transparency appears more frequently in the discourses of Portuguese respondents in keeping up with the problems surrounding sexual reassignment surgeries in the Portuguese public service [29].


*“(…) now in Coimbra, we don’t know. I know how few surgeries they are doing. I even did my augmentation mammoplasty there and everything went well, and I am satisfied with the result. I feel this validates the health service. However, doing my genital surgery there, I didn’t feel the slightest confidence in the service and didn’t want to risk it”.*
(3-T.F.PT.PL)

The “slowness of services” code is transversal across the participant responses and the greatest criticism presented to the medical model in addition to reproducing “normalization devices” in bodies and services derived from the slow pace of delivering those same services. In some cases, this slowness compounds the vulnerability of people, especially when some are not employed (which aggravates situations of vulnerability). These people live in such precarious situations that waiting times can pose a real obstacle to survival [25]. The following is an example:


*“(…) with a very long line, with abuses, medical arbitrariness, with very, let’s say, aesthetically negative surgical results, you can already find a series of complaints. (…) But one of the biggest complaints is really the waiting line. (…) It’s not for nothing that people go by other means, they won’t wait, it’s their life. Nobody wants to wait to be happy”.*
(21-L.M.BR.S)

In the “real life experience(s) and their questioning” code, the participant answers were diverse. If, on the one hand, some participants referred to real-life experience as a test, and, therefore, as a way of trying to evaluate their identity, others referred to this experience as important to their process of self-construction, while emphasizing how this was a positive experience for them because, at that moment, they were already experiencing and (re)constructing their identity.

Some respondents also problematized how any real-life experience should not be considered as proof to legitimize any identity. Some studies identify how real-life experiences should indeed be adopted with care in the sense of not adding more stigma to people who already face a marked process of social stigmatization [5,6,30]. The following is one example of this:


*“Sometimes people, and especially transsexual women who have not yet done any hormone therapy and cannot at all have a female aspect, are not physically prepared for this experience but are somehow forced to have it not as an experience but rather as a test and that is a situation that is packed with dangers and even from being physically on the street”.*
(6-R.F.PT.GL)

Regarding the “non-legitimation of identity by professionals” code, some participants mention that many experiences of health services can turn out negative or positive depending on the “good” will of the professionals, as health professionals are responsible for protecting the legitimacy of trans identities [10,25]. In this specific case, some responses report a lack of legitimacy of trans people’s identities among health professionals, who are themselves responsible for defining the limits of their construction of femininity or masculinity. This situation once again places the professional in the role of judging someone’s desires and identities, thereby devaluing the self-determination of the bodies and identities of trans people themselves, whilst the latter should be the protagonists of their own lives [6,10,31]. The following is an example:


*“But I’m interested in making the registration change and everything else, only we lose the will. It’s so many documents you have to take along and all the medical reports, you know. My word is not enough, you have to be the doctor, then…”.*
(18-B.F.BR.S)

Another code emerging from the data was what we called “the need for monitoring and not evaluation”. This code demonstrates how much participants feel evaluated (and on trial) in a process that should be more far more about follow-up and not identity evaluation. There were (bio)psychomedical paradigm shifts from one paradigm, which assumed identity could be evaluated, to a paradigm that believed it possible to measure the suffering of people who present non-conformity between sex and gender. Despite this paradigm shift, we are still unable to evaluate identity or measure suffering [6,32]. In fact, it is currently only possible to accompany people and inform them of the implications of their decisions, and hence the importance of recognizing the right to autonomy and management over their bodies and identities [5,6,33,34]. Some participants also mentioned that the (bio)psychomedical process focused on evaluating people and not on monitoring and following them up, especially after sexual reassignment processes (when occurring). The following represents an example of participant discourses on the “need for follow-up and not evaluation” code:


*“I think medical follow-up is of fundamental importance but not in the way it is approached today. In a pathological way, as if it were an illness, I don’t see it that way, but medical accompaniment is important, psychiatry, psychology, in short, are there to help us but not in this way of treating the person as sick”.*
(18-B.F.BR.S.)

Another code that emerged was the “lack of information and training”. The interviewees reported that from their experiences in health services, they could perceive the lack of information and training of professionals, which conditioned their interventions with trans people [25]. This lack of information and training also hindered the transmission of adequate information to trans people, causing some of them not to acquire information and training about their process. This lack of information and training is present both in Brazil and in Portugal. The following is but one example:


*“(…) doesn’t have a focus on trans health, much less on male trans, it doesn’t. So the trans man is still obliged to go and take care of himself in a gynaecologist. And suddenly a trans man arrives in front of her and she doesn’t know how to do it properly. But it’s the only option he has and even though he has a biologically feminine body that needs treatments directed to this area, he doesn’t have a professional prepared to deal with that, with this novelty, that there is a trans man with a feminine body, and from there, all kinds of moral, ethical, physical constraints happen to this man (…)”.*
(17-B.M.BR.CO)

Finally, and related to the previous “lack of information and training” code, some participants referred to the “need to create specialized services”, precisely because they perceived this gap when resorting to health services. In addition to the need for specialized services, there is also a need for the distribution of these services to various regions or urban centers, both in Portugal and Brazil. People who live in locations far from the large urban centers or regions where these services do exist, even if only precariously, encounter added difficulties in gaining access. Therefore, it is essential to create decentralized structures and improve/specialize the existing services. The following is an example:


*“At the level of specific care in Portugal, we have much to do and develop. (…) there is a process of adaptation that is not taking place and that also has to be done, both for health professionals to be prepared to receive trans people, and for trans persons themselves as the health care is not the same. A trans person cannot access general health care in the same way as a cis woman. There are specificities that have to be respected and trans persons also require educating in”.*
(3-T.F.PT.PL)

The contexts and the (bio)psychomedical diagnoses effectively establish divisions between those who want to modify their bodies (and the kinds of modifications) and those who do not. The medical model, and, consequently, the diagnosis as an official recognition of transsexualities, is so imbricated in the collective imaginary of trans people that they also delegitimize those who were not diagnosed as transsexuals. In this context, there lacks the appropriate recognition of self-determination with the medical model, consequently leading to health professionals who hold the institutional power of policing trans identities and bodies. In some cases, this policing is also carried out by some trans people, putting pressure on “beginners” to start the itinerary of transsexual body modification [5]. To better illustrate the “Experiences in health services” theme, the map and its codes are presented in Figure 3.

After the identification of the themes and throughout this process, we were able to analytically develop the following central organizer: “(bio)psychomedical interventions in the bodies and in the trans-experiences”. A thematic map of the analysis, with the themes and the central organizer of the analysis, is presented in Figure 4.

All the people interviewed in this study talked about these interventions and about the importance of them in the path of gender crossing, even if they have not had or do not wish to have sexual reassignment surgery. This gender trajectory reaches beyond genital surgery, even if the construction of many respondents’ identities extends beyond body modification (although not necessarily beyond surgery). In addition, the impact of these interventions on the construction of the subjects interviewed and trans people in general must be problematized in order to allow for an effective guarantee of access to healthcare for all people. To better illustrate the central organizer and the three analytical themes, a thematic map is presented in Figure 4.

## 4. Conclusions

The analysis of the discourses listed and systematized in this study allowed us to problematize the concept that the legitimization of (dis)identity(s) must focus on the self-definition and self-determination of people as opposed to attempts to legitimize any cishetero-elected identity, especially that determined by the biomedical model.

Through the responses of the participants, the impacts of “normalization devices” on their bodies, and, consequently, on their identities, also came to be questioned. This problematization stemmed not only from trans participants linked to activism but also from other participants who themselves reflected various contexts and spaces, including many of the questionings that arose precisely from people who referenced their contexts of (bio)psychomedical interventions.

The study also conveys evidence that some trans people feel that professionals do not legitimate their identities, and therefore, their health service experiences depend on the “good will” of professionals, as they were deemed the gatekeepers of the binary sex/gender system with protective power over the legitimacies of trans identities.

It is also important to point out that, in the context of criticizing the current medical model, sexual reassignment surgery may not be a desired goal for some transgender people [35] (for health reasons, the implications of surgical intervention, and fears over loss of pleasure, among others). In some cases, trans people undergo surgery so that they can adjust their sex to their gender. However, this desire incorporates another underlying factor: that such surgeries serve as a tool for repairing the recognition of their belonging to humanity. Instead of sexual reassignment surgery, for these people, the legal recognition of their sex and identity would be the most appropriate option for their physical and psychological well-being [15,36].

Currently, there is a historical opportunity to critically intervene in the medical discourses that regulate the lives of trans people. It remains a paradox that people who seek help and wish to initiate processes of body alteration (not necessarily of sex) must undergo diagnosis. It is not possible to separate the practice and terms of this diagnosis from the ethical context. However, it is possible to change the language used in order to shift the terms of the diagnosis, even if such linguistic changes need to be carried out according to ethical demands, which reside fundamentally in respect for the multiple experiential and expressive realities of transsexualities. In this way, it shall become possible to change the evaluation and assistance process so that biomedical entities (as well as the psychological positions uncritically submitted to a biomedical perspective) do not dictate “the truth” of someone’s wishes [32]. Thus, people might then be able to stand up to the institutions that continue to discipline and police what they insist on naming as the “deviations” from their bodies and/or their identities.

## Figures and Tables

**Figure 1 ijerph-19-00267-f001:**
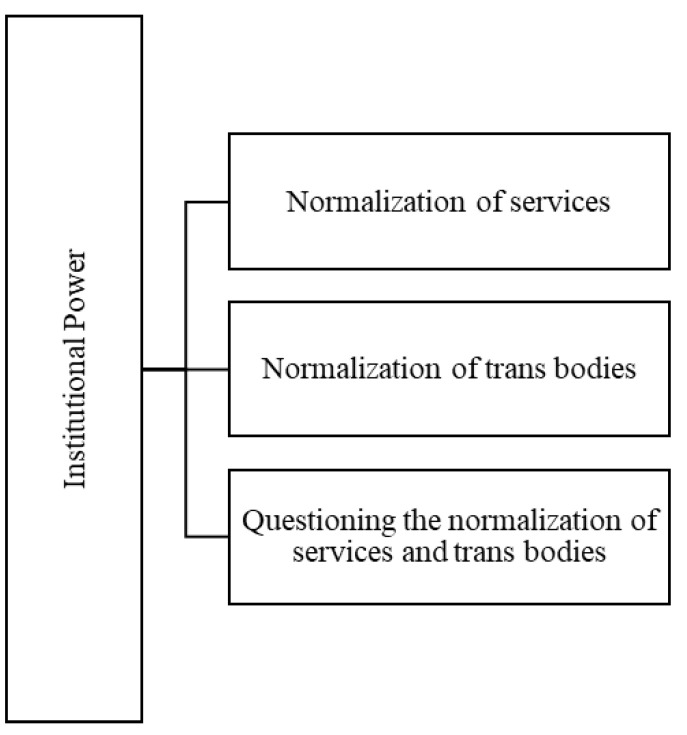
Map of the theme, “Institutional Power”.

**Figure 2 ijerph-19-00267-f002:**
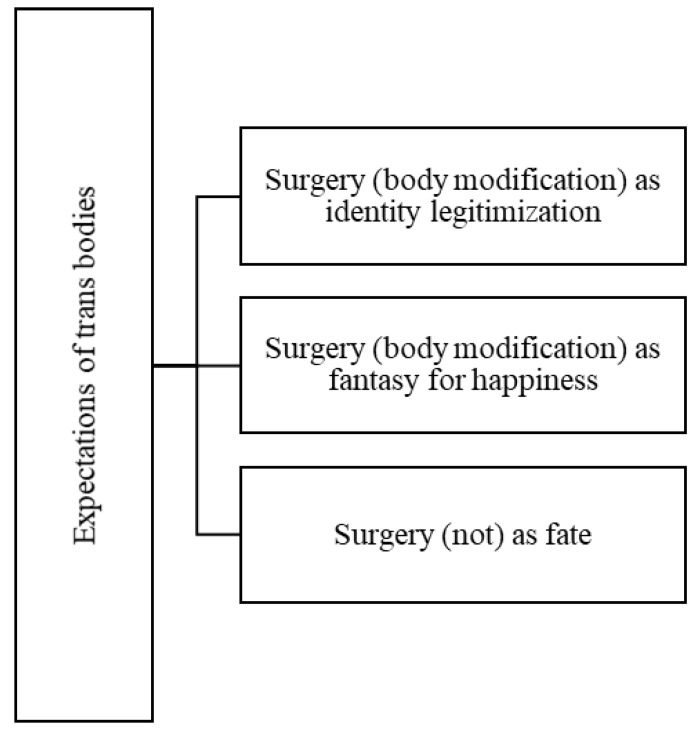
Map of the theme, “Expectations of trans bodies”.

**Figure 3 ijerph-19-00267-f003:**
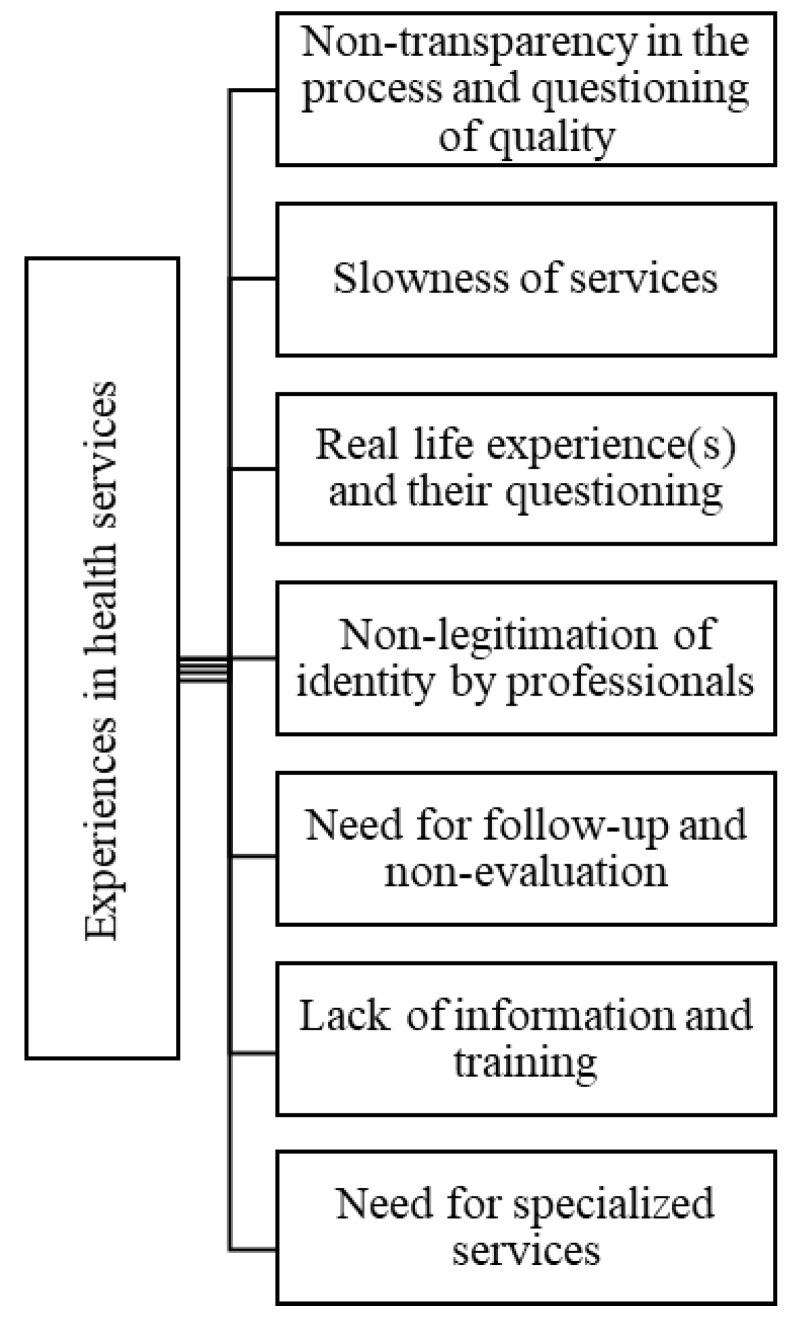
Map of the theme, “Experiences in health services”.

**Figure 4 ijerph-19-00267-f004:**
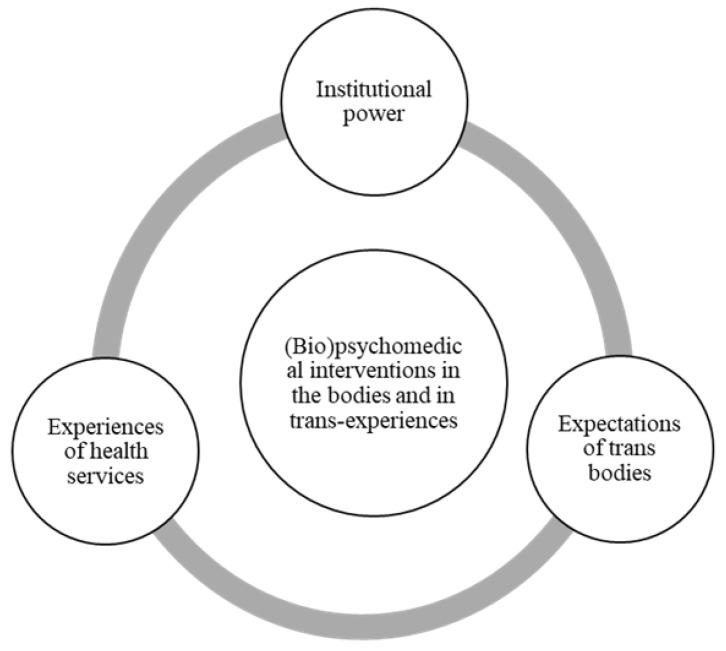
Thematic map of the analysis.

## Data Availability

Not available.

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
