# Peer review of "Psychomedical Interventions with Transgender People in Portugal and Brazil: A Critical Approach"

_ijerph, 2021, doi:10.3390/ijerph19010267_

Round 1

Reviewer 1 Report

This is an important and interesting study. Some minor comments and suggestions for improvement are below.

Abstract: This may be a cultural issue - should “transsexuals and transvestites” be re-labeled as “transgender” for publication purposes (though of course indicating if those are the terms used in data collection). Perhaps unnecessary given the definitions presented in the methods section, though these were a bit unclear.

Page 2: “Maintaining the pathology in these bodies involves maintaining the idea that these bodies will only become desirable, and may indeed wish to go through, processes of heteronormative…” Heteronormative what? There seems to be a word missing. Further, the rest of this paragraph appears to belong in the methods section, not here.

Methods paragraph 1: It’s a bit difficult to follow the differences in terminology presented here. Please revise this section to make it a bit more clear.

Study instrument section: Please expand on how the Portuguese and Brazilian instruments are different or similar.

Materials and methods section, last sentence: This is an unnecessary statement. Remove.

The term “normalization device” is used throughout. In context, it sounds as if these refer to procedures; however the word “device” makes it sound like a medical instrument. Please clarify.

While there is some discussion within each subheading of “results and discussion” there is not much by way of overall discussion, which would be helpful in understanding the data.

Author Response

Cover letter attched

Reviewer 2 Report

  1. This paper is well written and of particular interest to those concerned with the situation for transgender persons in Brazil and Portugal.
  2. I agree that surgery should not be the criterion for successful "transition".  Not only does it reinforce a pathological medical model but it overlooks the human potential for change in style/degree of transgender identification or desire.
  3.   It would be helpful to present a table crosstabulating transgender status with sexual orientation, even if no statistics are presented.
  4. It should be clarified if any of the papers' authors were participants.
  5. Page 5, second line in paragraph titled Results and Discussion, paradigm should be plural (paradigms).
  6. The FCT abbreviation should be spelled out in the funding paragraph, ideally with an address.
  7.   It seems there are about 13 quotes presented from 35 interviews.  I am concerned about this because author bias/theory might select quotes supporting the bias/theory rather than letting the theory be developed from the interviews.  I am not an expert in qualitative analysis so I hope that a reviewer so qualified might add more to this issue with specific recommendations.
  8.  The paper seems to ignore the theoretical possibility that a transgender identity might be very logical.  Suppose, for example, that you are a lesbian but your religion is opposed to it, so you might become transgender as a way of resolving that conflict.  A natal woman might come to her senses and realize that she's just not into menstruation, pregnancy, having children but she does want to be aggressive in a career, so identifying as a man and transitioning could solve her concerns.  I knew a professor once who had her ovaries and uterus removed because she didn't want to have to mess with menses and possible pregnancy; it wouldn't be too much further to transition to a male identity.  It made her free from societal prejudices about menstruation and from worry about any unwanted pregnancies or the need to get an abortion.  Future research ought to, IMHO, look into the natural logic of transgenderism and see to what extent transgender persons might "see" the logic, at least in hindsight. 

Author Response

Cover letter attached
